# Phase-based coordination of hippocampal and neocortical oscillations during human sleep

Roy Cox[1✉], Theodor Rüber[1,2,3], Bernhard P. Staresina[4] & Juergen Fell[1]

During sleep, new memories undergo a gradual transfer from hippocampal (HPC) to neocortical (NC) sites. Precisely timed neural oscillations are thought to mediate this sleep-dependent memory consolidation, but exactly how sleep oscillations instantiate the HPC-NC dialog remains elusive. Employing overnight invasive electroencephalography in ten neuro-surgical patients, we identified three broad classes of phase-based communication between HPC and lateral temporal NC. First, we observed interregional phase synchrony for non-rapid eye movement (NREM) spindles, and N2 and rapid eye movement (REM) theta activity. Second, we found asymmetrical N3 cross-frequency phase-amplitude coupling between HPC slow oscillations (SOs) and NC activity spanning the delta to high-gamma/ripple bands, but not in the opposite direction. Lastly, N2 theta and NREM spindle synchrony were themselves modulated by HPC SOs. These forms of interregional communication emphasize the role of HPC SOs in the HPC-NC dialog, and may offer a physiological basis for the sleep-dependent reorganization of mnemonic content.

[1] Department of Epileptology, University of Bonn, 53127 Bonn, Germany. [2] Epilepsy Center Frankfurt Rhine-Main, Department of Neurology, Goethe University Frankfurt, 60590 Frankfurt am Main, Germany. [3] Center for Personalized Translational Epilepsy Research (CePTER), Goethe University Frankfurt, 60590 Frankfurt am Main, Germany. [4] School of Psychology, University of Birmingham, B15 2TT Birmingham, UK. ✉email: roycox.roycox@gmail.com

A long-standing question in cognitive neuroscience asks how initially fragile episodic memories are transformed into lasting representations. Theoretical accounts postulate that this process involves a protracted transfer of memories from the hippocampus (HPC) to neocortical (NC) domains[1,2], with a large body of lesion[3,4], and neuroimaging[5,6] findings supporting this notion. One NC area of particular interest is the lateral temporal cortex, a convergence zone involved in long-term memory storage, representing higher-order visual, verbal, categorical, and semantic concepts[7–15].

Intriguingly, sleep leads to more stable and better-integrated episodic memories, suggesting a pivotal role for this brain state in the systems-level reorganization of memory traces. Specifically, it is thought that individual memory components, represented within different NC areas, are initially bound by HPC into an integrated whole, followed by a sleep-dependent HPC-NC dialogue to foster durable connections among the relevant NC sites[16–19]. Neural oscillations, especially non-rapid eye movement (NREM) neocortical slow oscillations (SOs; 0.5–1.5 Hz), thalamocortical sleep spindles (12.5–16 Hz), and hippocampal ripples (60–100 Hz[20–26]; note that human ripples are substantially slower than ~150–250 Hz rodent ripples[27,28]), are widely held to mediate this HPC-NC memory transfer and consolidation process[29–34], particularly given the presence of both SOs and spindles in HPC[22,24,35–37]. Moreover, various other spectral components exist in electrophysiological recordings of human sleep, with recent evidence suggesting potential roles for theta (4–8 Hz) in NREM[38,39] and rapid eye movement (REM)[40,41] memory processing, complicating the question of which oscillatory rhythms instantiate the HPC-NC dialog.

Oscillatory phase (i.e., the relative position along the oscillatory cycle) has a critical influence on neuronal excitability and activity[42], thereby offering a precise temporal scaffold for orchestrating neural processing within and across brain structures[43,44]. As such, oscillatory phase coordination between HPC and NC is a prime candidate mechanism for sleep-dependent information exchange between these areas. However, various forms of phase coupling may be distinguished, and phase-based HPC-NC interactions during sleep could be implemented in at least three (non-mutually exclusive) ways.

First, consistent oscillatory phase locking between brain regions at the same frequency is thought to enable effective communication between the underlying neuronal groups[45]. Phase synchrony during sleep has been reported between neocortical regions for various frequency bands[46], including the spindle[47,48] and gamma[49] ranges, and between HPC and prefrontal areas in the spindle range[50]. Whether similar phenomena exist between human HPC and non-frontal NC areas, for which frequency bands, and in which sleep stages, has not been examined.

Besides potential phase coupling within frequency bands, NREM sleep oscillations are also temporally organized across frequency bands. Such cross-frequency phase-amplitude coupling (PAC) is thought to enable brain communication across multiple spatiotemporal scales[51,52]. Local PAC among SOs, spindles, and ripples has been well characterized for various brain structures including HPC[21,22,24,47,53–57], and is considered a fundamental building block of memory consolidation theories[58]. However, local PAC exists for other frequency pairs[22], with SOs exerting particularly powerful drives not only over spindle and ripple activity, but also over delta[59], theta[60], and gamma[47,49] components. Extending the notion of local PAC to cross-regional interactions, the phase of a slower rhythm in one brain structure may modulate expression of faster activity at the other site[21,24,27], thus constituting a second potential form of HPC-NC communication.

Third, interregional phase synchronization within a frequency band might itself be modulated by the phase of a slower rhythm, as shown for the SO-phase-dependent coordination of spindle synchrony in neocortical networks[47]. Whether analogous SO-based modulation of phase synchronization exists between HPC and NC, and if so, for which frequency components, remains unexplored.

Here, we examined intracranial electrophysiological activity in a sample of 10 presurgical epilepsy patients during light NREM (N2), deep NREM (N3) and REM sleep. Specifically, we focused on HPC and lateral temporal cortex as a neocortical site relevant for long-term memory storage. Although sleep oscillations are not expressed uniformly across NC[61,62], findings of local SOs and spindles in virtually all of NC[35,36,53] make lateral temporal cortex a suitable site for studying HPC-NC interactions. We hypothesized that these areas exhibit interregional phase coordination, which could manifest in any or all of the three aforementioned forms of coupling. Given both the theoretical importance of nested SO–spindle–ripple activity[63], and inconclusive evidence regarding the directionality of HPC-NC coupling[35,37,50,64–68], we were particularly interested in whether HPC and NC SOs or spindles modulate faster activity at the other brain site, and if so, whether these effects are direction-dependent. Moreover, we considered a wide 0.5–200 Hz frequency range to allow potential identification of oscillatory communication lines outside the canonical SO–spindle–ripple framework. Using this approach, we identified several forms of sleep-based HPC-NC communication centered on SO, theta, and spindle activity, thereby offering a potential neurobiological substrate for sleep-dependent memory consolidation.

## Results

We analyzed overnight invasive electroencephalography (EEG) from the hippocampus (HPC) and lateral temporal neocortex (NC) in a sample of 10 epilepsy patients during N2, N3, and REM sleep. Polysomnography-based sleep architecture was in line with healthy sleep (Supplementary Table 1). Only intracranial contacts from the non-pathological hemisphere were used, as evidenced by clinical monitoring. Electrode locations are shown in Fig. 1 and Supplementary Table 2.

**Spindle and theta phase synchronization between hippocampus and neocortex.** Following inspection of raw traces with spectrograms (Supplementary Fig. 1) and power spectra (Supplementary Fig. 2), we evaluated whether, and to what degree, oscillatory signals in HPC and NC show phase coordination within frequency bands. Using the weighted phase lag index

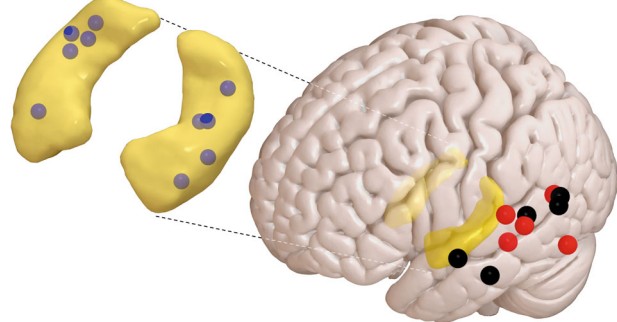

**Fig. 1 Group-level distribution of electrodes in hippocampus and on neocortical surface.** Outline of hippocampi in yellow, hippocampal depth electrodes in blue, subdural neocortical electrodes in red (left hemisphere) and black (flipped from right hemisphere). Electrode contacts not to scale. $N = 10$.

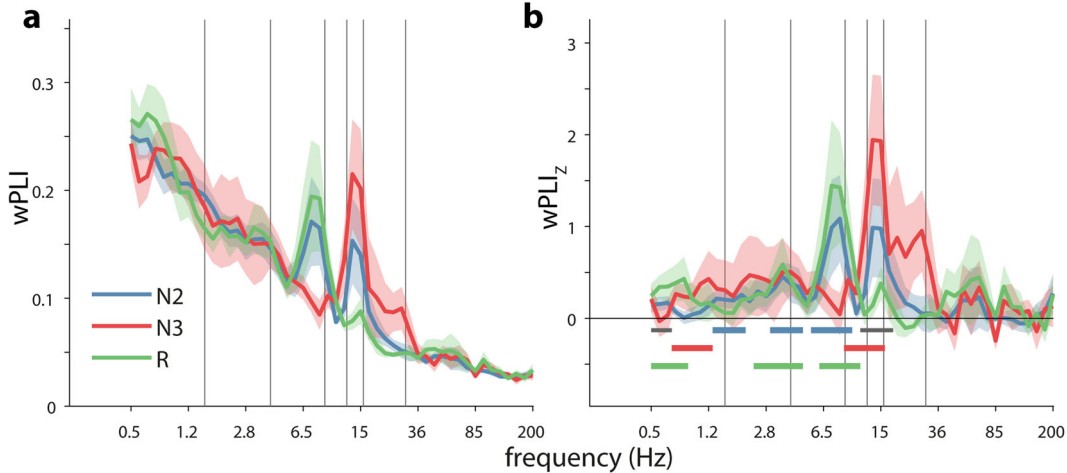

**Fig. 2 Phase synchrony between hippocampus and neocortex.** Group-level connectivity profiles for (unnormalized) wPLI (**a**) and (normalized) wPLI$_Z$ (**b**). Horizontal bars in (**b**) indicate above-chance connectivity (one-tailed cluster-based permutation versus zero; color: $P < 0.05$; gray: $P < 0.1$). Error shading: standard error of the mean across patients. Gray vertical lines at 1.5, 4, 9, 12.5, 16, and 30 Hz indicate approximate boundaries between SO, delta, theta, slow spindle, fast spindle, beta, and faster activity. $N = 10$.

(wPLI: a metric minimally sensitive to common neural sources[69]), we observed that raw wPLI showed a general decrease with frequency (Fig. 2a), with slower rhythms showing stronger synchrony than fast oscillations, as typically observed[70]. However, clear departures from this downward trend appeared for two frequency bands. First, phase synchrony enhancements appeared in the spindle range (peak frequency: 13.6 Hz) during N2 and especially N3 sleep, consistent with findings from other brain sites[47,50]. Second, an unexpected synchronization peak was observed in the theta range (peak frequency: 7.4 Hz) during N2 and REM.

To determine whether phase coupling in these or any other frequency bands was beyond chance levels, we z-scored raw wPLI values with respect to time-shifted surrogate distributions. This procedure essentially removed the downward trend, while retaining the aforementioned theta and spindle peaks (Fig. 2b). Comparing wPLI$_Z$ values to zero (one-tailed cluster-based permutation test; significant ranges indicated by colored bars at bottom of Fig. 2b) yielded several clusters of above-chance HPC-NC phase coordination. Considering frequencies from high to low, we observed reliable spindle synchronization during N3 (9.4–15.3 Hz, $P = 0.005$), and, at a more lenient threshold, during N2 (12.0–17.3 Hz, $P = 0.08$). Significant theta connectivity was seen for both N2 (5.8–9.4 Hz, $P = 0.03$) and REM (6.5–10.6 Hz, $P = 0.006$). Various control analyses showed that spindle and theta phase synchrony were not systematically related to power (Supplementary Fig. 3 and Supplementary Results). In addition, weaker, but significant, clusters were found for delta connectivity during N2 (3.1–4.5 Hz, $P = 0.03$) and REM (2.5–4.5 Hz, $P = 0.001$), while SO–range synchronization was found for N2 (1.3–1.9 Hz, $P = 0.03$; 0.5–0.6 Hz, $P = 0.09$), N3 (0.7–1.2 Hz, $P = 0.025$), and REM (0.5–0.8 Hz, $P = 0.009$).

Overall, these findings indicate a precise phase-based coordination between HPC and NC rhythms in several frequency bands, most clearly in the spindle and theta ranges, but also for delta and SO activity.

**Cross-frequency coupling of neocortical activity to hippocampal slow oscillations.** Next, we turned our attention to interactions between, rather than within, frequency bands. We quantified cross-frequency coupling using the debiased phase-amplitude coupling metric (dPAC: a metric correcting for potential non-sinusoidality of the phase-providing frequency[71]).

These values were further z-scored with respect to surrogate distributions. The resulting metric (dPAC$_Z$) signifies the degree to which activity at a faster frequency is non-uniformly distributed across the phase of a slower frequency.

Following analyses of local PAC within HPC and NC separately (Supplementary Results and Supplementary Figs. 4–6), we asked whether the oscillatory phase in one brain area could modulate faster activity in the other region. Importantly, such analyses enable directional inferences[72–74]. Assessing whether the phase of HPC rhythms coordinates faster activity in NC ("HPC-NC PAC"), we found that HPC SOs (0.5–1 Hz) robustly orchestrate the expression of faster activity in NC during N3 sleep (Fig. 3a). Specifically, distinct hotspots were found for modulated frequencies in the delta (maximum: 3.5 Hz), theta (6.5 Hz), beta/low-gamma (32 Hz), and high-gamma/ripple (85 Hz) ranges (white arrows in Fig. 3a). No systematic cross-regional modulation of neocortical activity by the hippocampal phase was observed during N2 or REM sleep. Interestingly, and in stark contrast to the robust modulation of NC activity by HPC SOs, the NC SO phase did not reliably coordinate faster HPC dynamics for any frequency band or sleep stage, nor did the phase of any other NC frequency modulate HPC activity ("NC-HPC PAC", Fig. 3b).

We further investigated this apparent asymmetry in how the N3 SO rhythm coordinates distant activity in various manners. First, we extracted individuals' dPAC$_Z$ values for the four visually apparent frequency pairs showing maximum HPC-NC group effects (white arrows in Fig. 3a), along with their opposite direction counterparts (gray arrows in Fig. 3b). Unsurprisingly, coupling strength for each selected frequency pair was significantly greater than zero for the HPC-NC direction (one-tailed Wilcoxon signed rank test with False Discovery Rate (FDR) correction, all $P_{corrected} < 0.02$). In contrast, no above-chance coupling was observed in the opposite NC-HPC direction (all $P_{uncorrected} > 0.31$). (Note that the preceding comparisons to zero are not independent of the cluster-based results of Fig. 3a, b, but are included for illustration and completeness.) Moreover, directional comparisons indicated that interregional PAC was systematically greater for HPC-NC vs. NC-HPC coupling for each frequency pair (two-tailed Wilcoxon signed rank test: all $P_{corrected} < 0.03$), as further illustrated in Fig. 3c. We also directly compared the full interregional coupling profiles of Fig. 3a and b, yielding a highly similar pattern of enhanced HPC-NC vs. NC-HPC coupling centered on the N3 SO-band

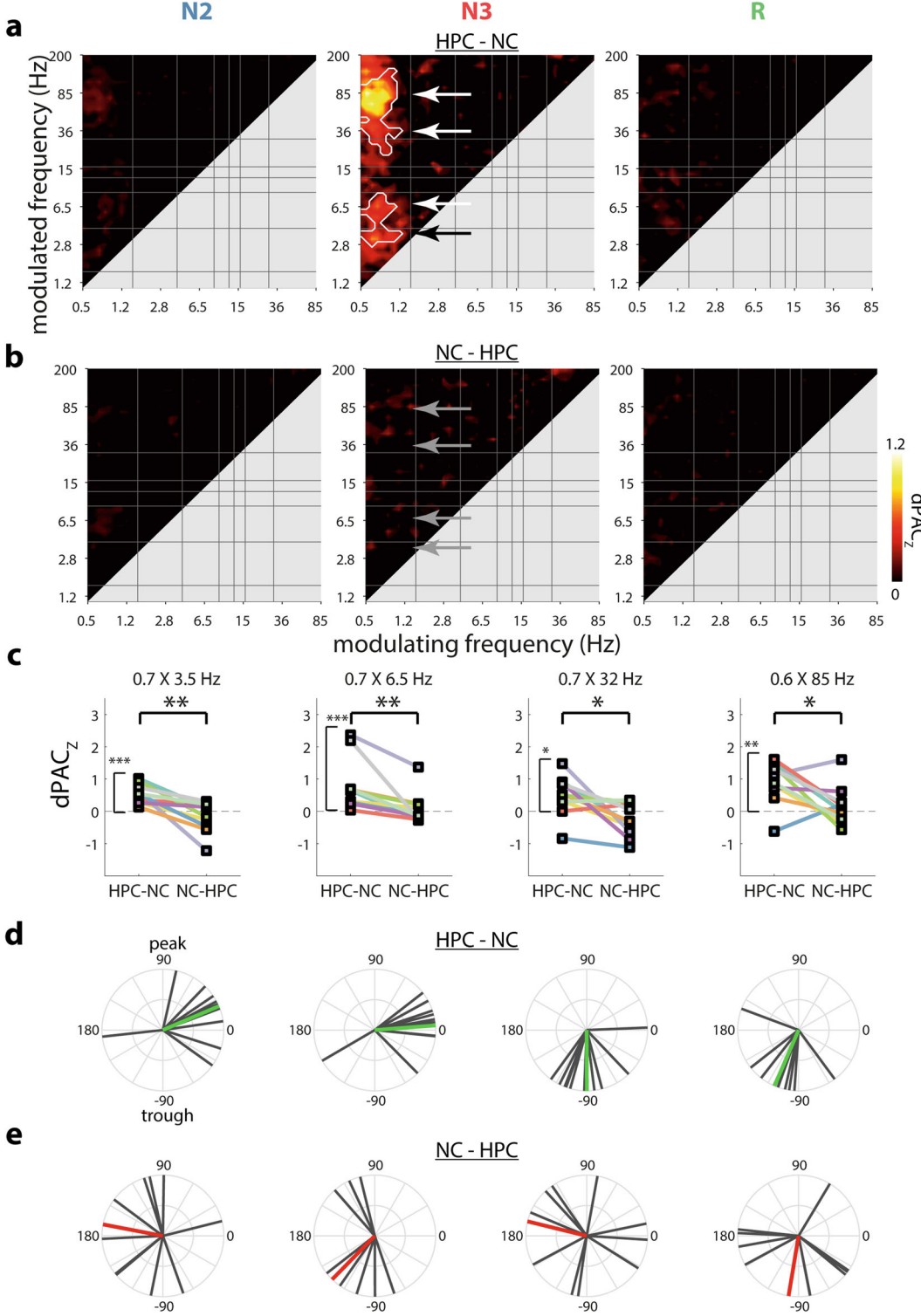

**Fig. 3 Cross-frequency coupling between hippocampus and neocortex.** Coupling strengths (dPAC$_Z$) for HPC-NC (**a**) and NC-HPC PAC (**b**). White outlines indicate clusters of significantly greater than zero coupling across patients (cluster-based permutation test). **c** Comparisons of HPC-NC and NC-HPC PAC for each SO-based N3 cluster (indicated in panels (**a**) and (**b**) with arrows). *$P < 0.05$, **$P < 0.01$, ***$P < 0.001$ (Wilcoxon signed rank test, uncorrected). **d**, **e** SO phase (with respect to sine wave) at which faster activity is maximally expressed across patients for HPC-NC (**d**) and NC-HPC (**e**) PAC. Colored lines indicate group averages, with green indicating significant ($P < 0.05$) deviations from uniformity, and red nonsignificance. $N = 10$.

(Supplementary Fig. 7). Of note, individual profiles of inter-regional coupling were highly consistent with these group-level findings (Supplementary Fig. 8).

Second, we considered, for the SO-based frequency pairs of N3, the precise phase at which distant fast activity was maximally

expressed. For HPC-NC PAC (Fig. 3d), phase distributions deviated substantially from uniformity for each of the four clusters (Rayleigh test for uniformity with FDR, all $P_{corrected} < 0.008$), indicating that fast NC activity is preferentially expressed at similar phases of the HPC SO across patients. Specifically, delta

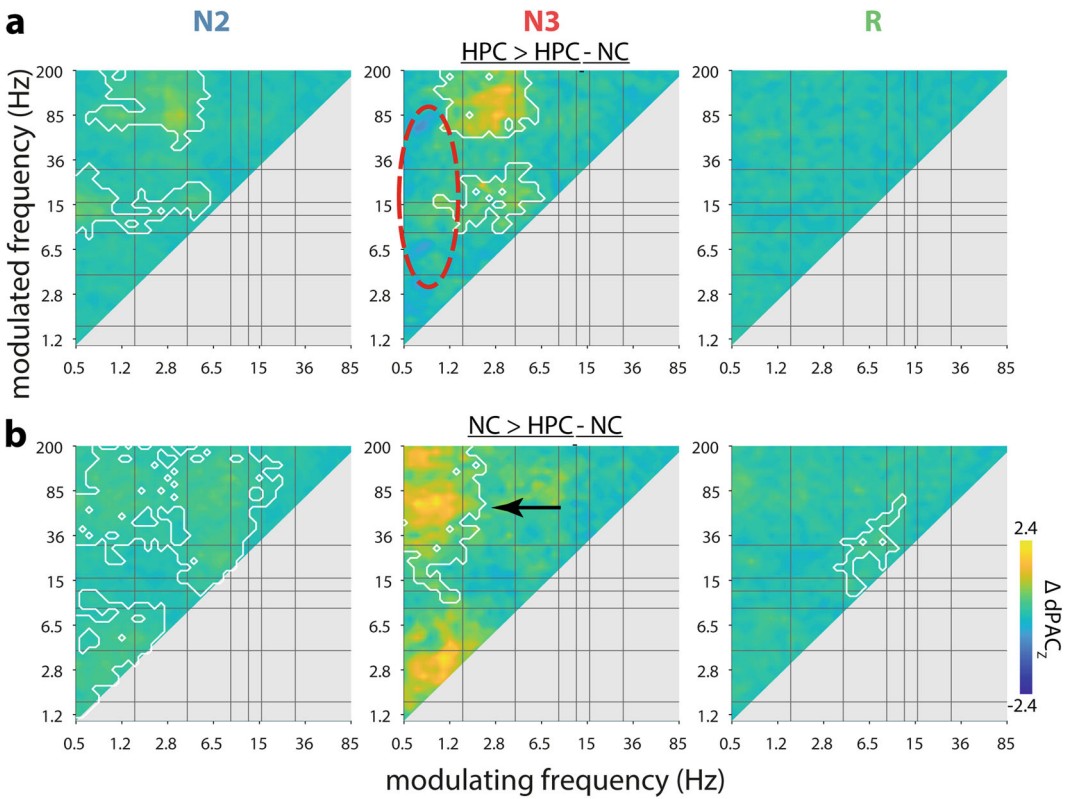

**Fig. 4 Differences between local and interregional cross-frequency coupling.** Coupling strength (dPAC$_Z$) differences for HPC versus HPC-NC (**a**) and NC versus HPC-NC (**b**). White outlines indicate clusters of significantly greater same-site than cross-site coupling (cluster-based permutation). No clusters with greater cross-site than same-site coupling were observed. N = 10.

and theta activity occurred around the negative-to-positive zero-crossing, while the low-gamma and high-gamma/ripple bands showed maximal activity in the SO trough (likely reflecting the physiological up-state[22,35]). In contrast, HPC fast activity was not consistently expressed in a particular phase range of the NC SO (theta: $P_{uncorrected} = 0.10$; all other $P_{uncorrected} > 0.31$; Fig. 3e), consistent with the lack of coupling reported in the previous paragraph.

Finally, we directly compared interregional HPC-NC PAC (as shown in Fig. 3a) with local PAC within each brain structure (as shown in Supplementary Fig. 4ab). For both HPC and NC, cross-frequency interactions were generally stronger within than between brain structures, as shown in Fig. 4. Specifically, the phase of HPC delta (1.5–4 Hz) organized spindle/beta and ripple activity more strongly within local HPC than in distant NC, most prominently during N3 (Fig. 4a). This delta-ripple effect is consistent with sharp wave-ripple complexes[22,75]. Interestingly, the N3 modulation of faster activity by the HPC SO was spared from these effects of enhanced local vs. interregional PAC (red oval in Fig. 4a), suggesting that HPC SOs are equally capable of modulating faster components in local and distant brain sites. In contrast, fast NC activity in the spindle-to-high-gamma bands during N3 was coordinated more robustly by the local NC SO than the HPC SO (black arrow in Fig. 4b). These observations could indicate that while fast NC activity is under the control of HPC SOs, local SOs still exert a stronger influence. While some local vs. interregional differences were also seen for N2 and REM, these effects should be interpreted cautiously, since no systematic interregional PAC was seen for these sleep stages (Fig. 3a).

Overall, these findings indicate that the HPC SO phase is capable of coordinating the expression of faster activity in NC regions during N3 sleep, whereas the reverse NC-HPC

modulation does not occur. Given these observations, as well as the within-frequency synchronization for theta and spindle rhythms (Fig. 2), an intriguing possibility is that SO rhythms also affect interregional phase synchronization. We address this question next.

**Modulation of interregional phase synchronization by hippocampal slow oscillations.** As a final potential form of phase-based HPC-NC communication, we asked whether within-frequency phase synchronization for faster frequencies could vary as a function of a slower oscillatory phase. We computed HPC-NC wPLI for each modulated frequency as a function of the phase (18 bins) of each slower frequency in either HPC or NC. We then determined a modulation index (MI)[76] for each frequency pair, indicating the degree to which wPLI values are non-uniformly distributed across the cycle of a slower frequency, and further normalized MI with respect to surrogate distributions. (Due to methodological considerations related to data length, two patients were excluded from N3 analyses.)

Intriguingly, these analyses revealed a strong organizing influence from the HPC SO on interregional phase synchronization (Fig. 5a). Specifically, HPC-NC theta synchrony was reliably modulated by HPC SOs during N2, whereas spindle synchrony was coordinated by both N2 and N3 HPC SOs, similar to scalp findings[47]. Note that these theta and spindle effects overlap well with the frequency bands showing interregional synchrony in Fig. 2. No other frequency bands showed synchronization modulations in relation to the phase of any other HPC oscillation, for neither NREM nor REM sleep. Moreover, we observed no statistically reliable modulation of phase synchronization by the NC phase, neither for SOs nor other frequency bands (Fig. 5b),

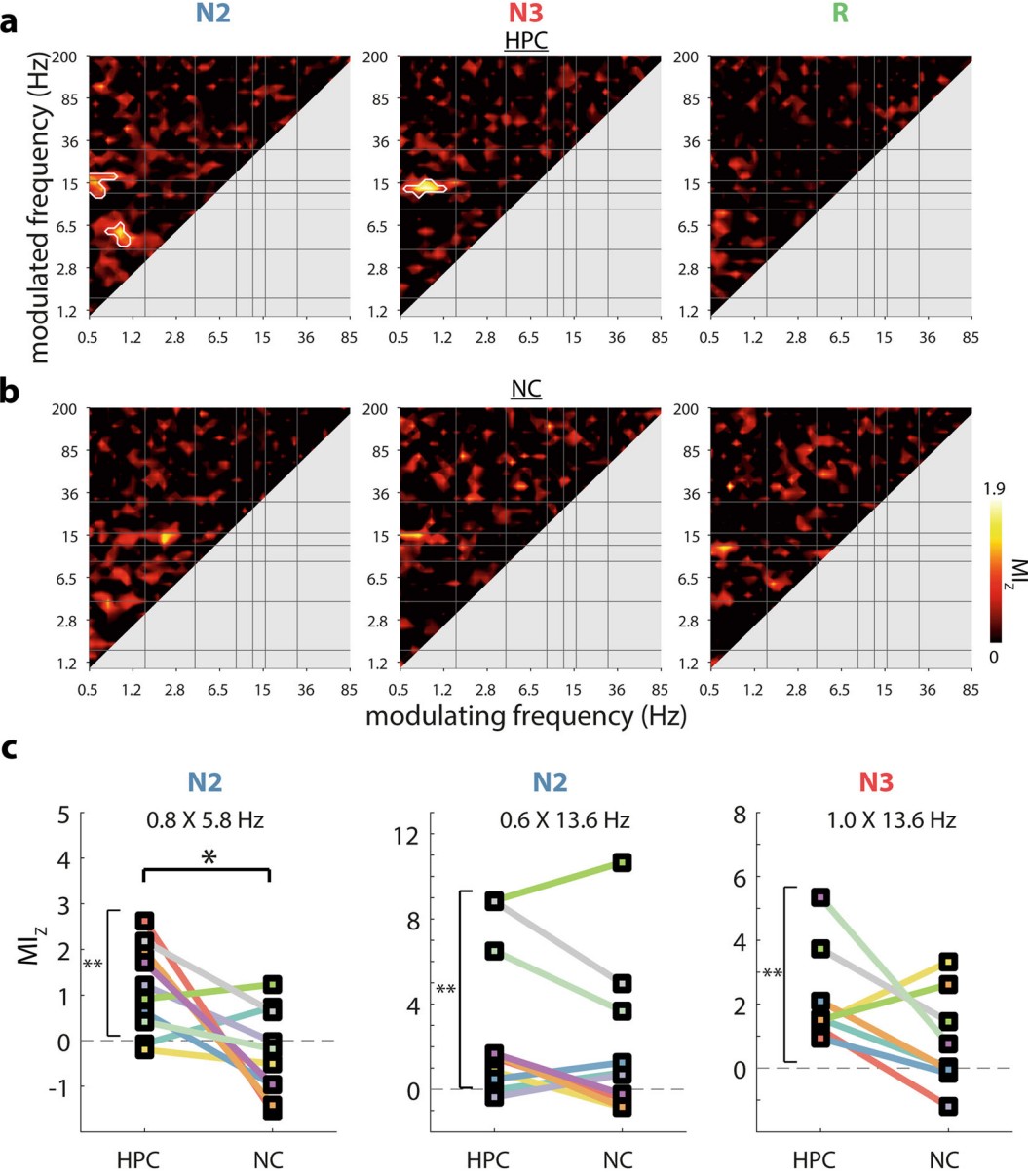

**Fig. 5 Cross-frequency modulation of phase synchronization between hippocampus and neocortex.** Normalized modulation indices ($MI_Z$) for phase of HPC (**a**) and NC (**b**). White outlines indicate clusters of significantly greater than zero modulation across patients (cluster-based permutation test). **c** Comparisons of HPC- and NC-phase-dependent modulation of phase synchronization for each SO-based cluster. *$P < 0.05$, **$P < 0.01$, ***$P < 0.001$ (Wilcoxon signed rank test, uncorrected). $N = 8$ for N3, $N = 10$ for N2 and REM.

although we note that faint hotspots suggestive of SO-spindle effects were visually apparent.

Similar to our approach for cross-regional PAC, we extracted patients' $MI_Z$ values for each of the three significant clusters of Fig. 5a. As visualized in Fig. 5c, these were all (trivially) significantly greater than zero (one-tailed Wilcoxon signed rank test with FDR, all $P_{corrected} < 0.005$), whereas their counterparts in the other brain region did not differ reliably from zero (N2 spindle: $P_{uncorrected} = 0.10$; other $P_{uncorrected} > 0.15$). (We again note the dependence of these effects on the cluster-based results of Fig. 5a, b.) Direct comparisons between HPC- and NC-based modulation of phase synchronization for these frequency pairs showed indications for greater HPC vs. NC influence of N2 SOs on theta synchrony (two-tailed Wilcoxon signed rank test with FDR, $P_{corrected} = 0.08$, $P_{uncorrected} = 0.03$). In contrast, while spindle synchrony also appeared to be more robustly modulated by the phase of HPC rather than NC SOs (Fig. 5c), these effects

did not reach significance (both $P_{uncorrected} = 0.11$). Finally, we directly compared the full HPC and NC profiles of Fig. 5a, b, but no significant clusters emerged (Supplementary Fig. 9).

In sum, the phase of SO activity orchestrates interregional phase synchronization in both the theta and spindle frequency bands, with HPC SOs having a particularly strong impact on theta synchrony. These findings establish another major form of phase-based HPC-NC coordination, potentially contributing to systems-level memory reorganization.

## Discussion

Communication between the hippocampus and neocortex during sleep is considered a cornerstone of theories of memory consolidation, but exactly how these interactions are instantiated in the human brain has remained unclear. In line with the notion that oscillatory phase is critically involved in binding distant but functionally related neural populations[43], we observed systematic

(i) within-frequency phase synchronization, (ii) cross-frequency phase-amplitude coupling, and (iii) cross-frequency modulation of within-frequency phase synchronization, thereby uncovering several previously unknown modes of interregional HPC-NC communication. A particularly prominent role emerged for HPC SOs, coordinating both the expression of, and synchronization with, faster NC activity.

As a first major form of phase-based interregional communication, we observed pronounced within-frequency theta and spindle phase synchronization between HPC and NC (Fig. 2), thus reflecting precise oscillatory coordination on a cycle-by-cycle basis for these frequency bands. Consistent phase relations between brain areas affect the relative timing of neuronal spikes, thereby enabling communication and plasticity[45,77].

Sleep spindles are closely tied to memory and plasticity[30,31,78,79], and show widespread phase synchronization in neocortical networks[47]. Here, we extend these observations of NREM spindle synchrony to include dynamics between HPC and lateral temporal cortex. These findings are consistent with observations of spindle-based communication between HPC and prefrontal areas[50]. Combined, these observations indicate a precise coordination of spindle activity across HPC and distributed neocortical areas, offering a potential mechanism for the reactivation of distributed memory traces, and thereby contributing to NREM-dependent memory consolidation.

Surprisingly, similar observations of HPC-NC phase synchrony were made for N2 and REM theta. These findings may offer a physiological basis for recent work demonstrating a role for NREM theta in memory consolidation[38,39]. In contrast, interregional HPC-NC theta synchrony during REM sleep could form a neurobiological basis for associations between REM theta and the regulation and consolidation of emotional content[40,41]. Combined with similar findings of REM theta connectivity between prefrontal and cingulate areas[80], theta rhythms thus appear to be coordinated across widespread brain areas. Of note, the observed REM theta synchrony contrasts with a study reporting no REM theta coherence between HPC and NC[81]. However, that observation was based on only two patients, providing limited opportunities to detect effects that may not be present in all individuals, as we also observed (e.g., Supplementary Fig. 3a). Of note, NREM synchrony for both aforementioned frequency bands varied with sleep depth. While the reason for differential connectivity during N2 and N3 is unclear, these findings underscore the need to consider these sleep stages separately.

We also observed phase-based connectivity in the SO (0.5–1.5 Hz) and delta (1.5–4.5 Hz) ranges. SO-band synchrony was found in all sleep stages, consistent with the existence of REM SOs[82], whereas delta synchrony was limited to N2 and REM. Synchronized SO activity might be expected given the presence of SOs in both NC and HPC[22,24,35,37]. In contrast, the origin of the delta effect is less clear, and could reflect a continuum with the SO band, an independent delta-band oscillation, HPC sharp wave activity[22,75], or any combination of these. While statistically significant, SO- and delta-based connectivity were much lower relative to the theta and spindle effects. This suggests that slow components in HPC and NC show relatively variable phase relations[37,50], which is further consistent with the local expression of SOs[35,53]. However, we note that phase synchrony profiles differed between individuals, with SO and delta synchronization peaks sometimes observed on an individual basis (Supplementary Fig. 3a–c).

As a second major form of oscillatory HPC-NC coordination, we observed systematic interregional cross-frequency phase-amplitude coupling. These effects were restricted to a governing role of HPC SOs over NC activity spanning the delta, theta, low

gamma, and high-gamma/ripple ranges (Fig. 3a). The opposite pattern, whereby the phase of NC oscillations coordinates HPC activity, for either SOs or other frequencies, was not seen (Fig. 3b, c). Similarly, the preferred SO phase at which faster activity was expressed was highly consistent across patients for HPC-NC, but not NC-HPC PAC (Fig. 3d, e). Of note, this asymmetry is consistent with the notion of independent HPC and NC SO dynamics, as suggested by relatively weak SO phase synchrony.

Although our metric of interregional PAC does not contain directional information per se, it is widely assumed that it is the phase of the slower frequency that modulates faster activity, rather than the other way around[51,52]. Importantly, contrasting interregional PAC calculated in opposite orders may be used to infer directional influences[72–74]. While SOs and their coordination of faster activity are typically viewed as NC phenomena (Supplementary Fig. 4b), similar dynamics within HPC are now well established (Supplementary Fig. 4b)[22,24,35,37,64,83,84]. As such, our findings suggest a driving force of HPC SOs on NC activity, co-determining NC activity in various faster frequency bands. These effects may stem from surges of local activity associated with HPC up states being transmitted to post-synaptic targets and eventually reaching NC. Indeed, while faster activity was typically modulated more strongly by local than distant slower rhythms, HPC SO activity coordinated local and NC faster activity to similar extents (Fig. 4a), potentially fostering more efficient HPC-NC information exchange.

We did not observe systematic cross-regional HPC-NC PAC for modulating rhythms beyond SOs, although we did find such examples on an individual basis (e.g., HPC theta modulating NC beta/gamma/high-gamma activity, Supplementary Fig. 8a, p7, N2). This general lack of HPC-NC PAC beyond SOs is noteworthy given that many additional frequency pairs were coupled locally in HPC and NC (Supplementary Fig. 4). These findings indicate that cross-regional and local PAC are at least partially dissociated, which is further supported by the observed asymmetry between HPC-NC and NC-HPC PAC. Importantly, these findings also alleviate concerns that cross-regional PAC is due to volume conduction, whereby modulating, modulated, or both signal components primarily reflect activity from the other brain site.

The lack of systematic NC-HPC PAC in our data may appear at odds with previous observations of NC-HPC SO-spindle[24], SO-ripple[85], and spindle-ripple coupling[21,24,27], which could be related to several factors. First, previous work often assessed NC activity with non-invasive scalp electrodes that aggregate activity over large spatial domains, thus reflecting common signals with relatively powerful drives. In contrast, the localized NC activity we considered here constitutes only a tiny fraction of all NC activity and may therefore exert a more limited influence on HPC activity (also see relative dissociation of scalp and NC signals in Supplementary Fig. 1). Second, and somewhat related, it is possible that modulation of HPC activity by NC SOs only becomes apparent when considering frontal regions with high SO densities, amplitudes, and synchrony (e.g., refs. [62,85]). Indeed, interregional spindle synchrony is modulated by the phase of frontal, but not central or parietal, SOs[47], raising the possibility that SO activity recorded from lateral temporal areas is insufficient to impact HPC dynamics. We return to the issue of the NC recording site below.

The third and final form of oscillatory HPC-NC interaction we observed was the modulation of within-frequency phase synchronization by the phase of slower rhythms. Most prominently, the HPC SO phase had a robust influence on the degree of N2 theta and NREM spindle synchrony (Fig. 5a), matching the sleep stages where these forms of synchrony were apparent (Fig. 2).

The gating of spindle synchrony by HPC SOs is highly consistent with similar observations of SO-modulated spindle synchrony in scalp data[47], and compatible with findings of enhanced HPC-prefrontal spindle synchrony for spindles coupled vs. uncoupled to frontal SOs[50]. While we did not see unambiguous evidence that NC SOs impose a similar modulation on spindle synchrony, modulation strengths also did not differ reliably between HPC and NC. Hence, strong conclusions regarding whether HPC or NC SOs most effectively affect spindle synchrony are presently not warranted.

In stark contrast, N2 theta synchrony depended to a greater extent on HPC than NC SOs. Intriguingly, this effect appears to be separate from the enhanced HPC-NC vs. NC-HPC modulation of theta amplitude by SOs, which occurred in N3 rather than N2. While the reason for this dissociation is unclear, both effects are in agreement that interregional theta dynamics are modulated most effectively by HPC rather than NC SOs.

Our findings add to the debate on the directionality of HPC-NC dynamics during sleep. Prior empirical evidence has been mixed, pointing towards HPC-NC[66,85], NC-HPC[28,35,65], or more elaborate bidirectional paths[50,67,68]. Both our findings of HPC SOs coordinating NC faster activity but not vice versa, and of stronger modulations of theta phase synchrony by HPC SOs than NC SOs, are most consistent with the notion of HPC to NC directionality, as suggested by classical theoretical[16] and computational[86] models. Based on these results, we propose that HPC SOs may influence plasticity not only within HPC but also in NC circuits[64]. We emphasize, however, that observed directionality varies importantly with the precise electrophysiological phenomenon under consideration, and as mentioned, may also depend on the particular NC regions examined.

We considered communication between HPC and lateral temporal cortex, a neocortical region important for long-term memory storage[7,10,13,14]. More fundamentally, dominant sleep consolidation theories posit that the HPC-NC dialogue engages all NC areas involved in representing episodic memory components[16–19], which for lateral temporal cortex could constitute higher-order visual, verbal, and semantic concepts[8,9,11,12,15]. While our patient sample offered much wider NC electrode coverage, we deliberately studied a relatively circumscribed NC area (Fig. 1, Supplementary Table 2) to avoid confounding influences related to the non-homogeneous spatial expression of SOs and spindles[35,61,62,87]. Hence, we reiterate that the observed forms of communication, and their directionality in particular, may be specific to lateral temporal areas. At the same time, observations of local SOs and local spindles in widespread NC areas[35,36,53] suggest that the reported forms of phase coordination may apply more broadly to other NC sites.

Although generalizing from epileptic to healthy populations poses a risk, sleep architecture was in line with healthy sleep (Supplementary Table 1). Moreover, we employed a rigorous artifact rejection protocol, and only considered electrodes on the non-pathological side, making it unlikely our results are due to epileptiform activity. In the present approach, measures of oscillatory coordination were calculated over continuous data. This contrasts with discrete approaches where analyses are contingent on the presence of specific waveforms. Given that our approach identified various expected phenomena of local PAC (e.g., SO-spindle, spindle-ripple; Supplementary Figs. 4 and 6), we do not believe this methodological choice poses a major concern. Nonetheless, PAC metrics capture both true oscillatory interactions and features related to waveform shape[88], which may have contributed to our results. Although we believe that the specific patterns of results (e.g., asymmetrical HPC-NC modulation limited to the SO-band) are most parsimoniously explained by interacting oscillations, future work should

scrutinize individual waveforms to fully understand the origin of each of the observed effects.

In summary, the present observations establish an important prerequisite for memory consolidation theories postulating a sleep-dependent HPC-NC dialog[16–19]. More specifically, the identified forms of phase coordination draw attention not only to SOs and spindles, but also to theta activity. Furthermore, the asymmetrical coordination of NC activity and HPC-NC phase synchronization by HPC SOs suggests that HPC may play a larger orchestrating role in information exchange during sleep than previously thought. Overall, these findings refine our knowledge of human HPC-NC interactions and offer new opportunities to understand the determinants of sleep-dependent memory consolidation in health and disease.

## Methods

**Participants**. We analyzed archival electrophysiological sleep data in a sample of 10 (6 male) patients suffering from pharmaco-resistant epilepsy (age: 36.6 ± 14.8 yrs, range: 22–62). This sample overlaps with ones reported previously[22,24,65,89]. Local aspects of the HPC data are described in detail elsewhere[22], but are summarily included here both because of different patient and electrode inclusion criteria and to provide a comprehensive perspective on HPC-NC oscillations. Patients had been epileptic for 22.5 ± 11.0 yrs (range: 10–49) and were receiving anticonvulsive medication at the moment of recording. All patients gave informed consent, the study was conducted according to the Declaration of Helsinki, and was approved by the ethics committee of the Medical Faculty of the University of Bonn.

**Data acquisition**. Electrophysiological monitoring was performed with a combination of depth and subdural strip/grid electrodes. HPC depth electrodes (AD-Tech, Racine, WI, USA) containing 8–10 cylindrical platinum-iridium contacts (length: 1.6 mm; diameter: 1.3 mm; center-to-center inter-contact distance: 4.5 mm) were stereotactically implanted. Implantations were done either bilaterally ($n = 7$) or unilaterally ($n = 3$), and either along the longitudinal HPC axis via the occipital lobe ($n = 9$) or along a medial-lateral axis via temporal cortex ($n = 1$). Stainless steel subdural strip/grid electrodes were of variable size with contact diameters of 4 mm and center-to-center spacing of 10 mm, and placed over various neocortical areas according to clinical criteria. Anatomical labels of each electrode were determined based on pre- and post-implantation magnetic resonance image (MRI) scans by an experienced physician (TR), as described previously[22].

A single gray matter HPC electrode and a single NC electrode from lateral temporal cortex were selected for each patient. As reported previously for HPC, and here additionally seen for NC, within-patient spectral profiles varied between adjacent contacts. Following previous approaches[24] and the hypothesized central role of spindles in HPC-NC communication, the contact with highest NREM spindle power was chosen at both brain sites. MNI electrode locations for each patient are indicated in Supplementary Table 2, and were visualized using Surf Ice (https://www.nitrc.org/projects/surfice/) to generate Fig. 1. For HPC, fast spindle peaks were visible for all patients. For NC, 7 of 10 patients showed fast spindle peaks, one showed a slow spindle peak, and two did not exhibit noticeable spindle peaks. Additional non-invasive signals were recorded from the scalp (Cz, C3, C4, Oz, A1, A2; plus T5 and T6 in eight patients), the outer canthi of the eyes for electrooculography (EOG), and chin for electromyography (EMG). All signals were sampled at 1 kHz (Stellate GmbH, Munich, Germany) with hardware low- and high-pass filters at 0.01 and 300 Hz, respectively, using an average-mastoid reference. Offline sleep scoring was done in 20 s epochs based on scalp EEG, EOG, and EMG signals in accordance with Rechtschaffen and Kales criteria[90]. Stages S3 and S4 were combined into a single N3 stage following the more recent criteria of the American Academy of Sleep Medicine[91].

**Preprocessing and artifact rejection**. All data processing and analysis was performed in Matlab R2018a (the Mathworks, Natick, MA), using custom routines, EEGLAB[92], and CircStat[93] functionality. Preprocessing and artifact rejection details are identical to our previous report[22]. Briefly, data were high-pass (0.3 Hz) and notch (50 Hz and harmonics up to 300 Hz) filtered, and channel-specific thresholds ($z$-score > 6) of signal gradient and high-frequency (>250 Hz) activity were applied to detect and exclude epileptogenic activity. Artifact-free data "trials" of at least 3 s were kept for subsequent processing, resulting in a total of 78.1 ± 30.8 (N2), 21.7 ± 17.8 (N3), and 44.5 ± 23.7 min (REM) of usable data. We note that the relatively modest amount of remaining data primarily reflects our highly conservative artifact rejection approach, which was applied across many more channels than the ones included in the present study[22].

**Time-frequency decomposition**. Data were decomposed with a family of complex Morlet wavelets. Each trial was extended with 5 s on either side to minimize edge artifacts. Wavelets were defined in terms of desired temporal

resolution according to:

$$\text{wavelet} = e^{i2\pi t f} * e^{-4\ln(2)t^2/h^2} \tag{1}$$

where $i$ is the imaginary operator, $t$ is time in seconds, $f$ is frequency (50 logarithmically spaced frequencies between 0.5 and 200 Hz), $ln$ is the natural logarithm, and $h$ is temporal resolution (full-width at half-maximum; FWHM) in seconds[94]. We set $h$ to be logarithmically spaced between 3 s (at 0.5 Hz) and 0.025 s (at 200 Hz), resulting in FWHM spectral resolutions of 0.3 and 35 Hz, respectively. Trial padding was trimmed from the convolution result, which was subsequently downsampled by a factor four to reduce the amount of data. We normalized phase-based metrics using time-shifted surrogate approaches (see the "Surrogate Construction" section). To make surrogate distributions independent of variable numbers and durations of trials, we first concatenated the convolution result of all trials of a given sleep stage, and then segmented them into 60-s fragments (discarding the final, incomplete segment).

**Phase synchrony**. To assess within-frequency phase synchrony, we used the weighted phase lag index (wPLI)[69], a measure of phase synchrony that de-weights zero phase (and antiphase) connectivity. For every 60-s segment and frequency band, raw wPLI between seed channel $j$ (HPC) and target channel $k$ (NC) was calculated as:

$$\text{wPLI}_{jk} = \frac{\left| \frac{1}{n}\sum_{t=1}^{n} \left| imag(S_{jkt}) \right| sgn(imag(S_{jkt})) \right|}{\frac{1}{n}\sum_{t=1}^{n} \left| imag(S_{jkt}) \right|} \tag{2}$$

where $imag$ indicates the imaginary part, $S_{jkt}$ is the cross-spectral density between signals $j$ and $k$ at sample $t$, and $sgn$ indicates the sign. We further created a normalized version of this metric (wPLI$_Z$) using a surrogate approach (see Surrogate Construction). We used the median to further aggregate wPLI and wPLI$_Z$ values across data segments.

**Cross-frequency phase-amplitude coupling**. For every 60 s segment, PAC was determined between all pairs of modulating frequency $f1$ and modulated frequency $f2$, where $f2 > 2*f1$. We employed an adaptation of the mean vector length method[95] that adjusts for possible bias stemming from non-sinusoidal shapes of $f1$ and associated non-uniform phase distributions[71]. Specifically, complex-valued debiased phase-amplitude coupling (dPAC) was calculated as:

$$\text{dPAC} = \frac{1}{n}\sum_{t=1}^{n} (amp_{f2}(t) * e^{i\varphi_{f1}(t)} - B) \tag{3}$$

where $i$ is the imaginary operator, $t$ time, $amp_{f2}(t)$ is the magnitude of the convolution result, or amplitude, of $f2$, $\varphi_{f1}(t)$ is the phase of $f1$, and $B$ is the mean phase bias:

$$B = \frac{1}{n}\sum_{t=1}^{n} e^{i\varphi_{f1}(t)} \tag{4}$$

For same-site PAC (i.e., within HPC or within NC) $\varphi_{f1}$ and $amp_{f2}$ stemmed from the same electrode, whereas cross-site PAC used phase information from one brain structure and amplitude information from the other. Raw coupling strength (i.e., the degree to which the $f2$ amplitude is non-uniformly distributed over $f1$ phases) was defined as the magnitude (i.e., length) of the mean complex vector. We further created a normalized version of this metric (dPAC$_Z$) using a surrogate approach (see Surrogate Construction). We used the median to further aggregate dPAC$_Z$ values across data segments.

**Cross-frequency modulation of phase synchrony**. A modulation index (MI) was computed between all pairs of modulating frequency $f1$ and modulated frequency $f2$, where $f2 > 2*f1$. For each frequency $f1$, samples were binned according to phase $\varphi_{f1}$ (18 bins), and wPLI was calculated for each bin $b$ and frequency $f2$ following Eq. (2). Segment-averaged wPLI values were then used to calculate raw MI as:

$$\text{MI} = \left| \frac{1}{n}\sum_{b=1}^{n} \left( \text{wPLI}_{f2,b} * (e^{i\varphi_b}) \right) \right| \tag{5}$$

where $b$ is the bin number, $n$ is the number of bins (18), and $\varphi_b$ is the phase at each bin center. This calculation was performed separately for HPC and NC $f1$ phases. Note that MI was calculated across all available segments rather than per 60 s segment (as for wPLI and dPAC) because segment-wise MI estimates proved unstable. We further created a normalized version of this metric (MI$_Z$) using a surrogate approach (see Surrogate Construction).

**Surrogate construction**. For phase synchrony, surrogates were constructed per 60 s segment and frequency by repeatedly ($n = 100$) time shifting the phase time series of the seed channel by a random amount between 1 and 59 s, and recalculating wPLI for each iteration. Similarly, for cross-frequency phase-amplitude coupling, we constructed a surrogate distribution of coupling strengths per 60 s segment, frequency pair, and same/cross-site condition, by repeatedly ($n = 100$) time shifting the $f1$ phase time series with respect to the $f2$ amplitude time series, recalculating

the mean vector length (dPAC) for each iteration. For the cross-frequency modulation of phase synchrony, surrogate distributions were constructed by repeatedly ($n = 100$) shuffling the pairing of $f1$ and $f2$ 60 s phase segments (disallowing pairings where individual segments were unaltered), and recalculating MI across segments for each iteration, rather than per segment. Since the number of unique segment pairings depends on the number of available segments, two patients were excluded from N3 analyses.

Note that time shifting accounts for non-stationarities in the data, and is a more conservative approach than fully scrambling time series, which may result in spurious effects[96]. For each metric, the surrogate distribution was used to z-score the raw values. Thus, the z-scored measures (wPLI$_Z$, dPAC$_Z$, MI$_Z$) indicate how far, in terms of standard deviations, the observed coupling estimate is removed from the average coupling estimate under the null hypothesis of no coupling.

**Statistics and reproducibility**. Statistical analyses were performed at both the group (wPLI$_Z$, dPAC$_Z$, MI$_Z$) and individual (dPAC$_Z$) levels. wPLI$_Z$, dPAC$_Z$, and MI$_Z$ were assessed using cluster-based permutation tests[97] with a *clusteralpha* value of 0.1. We used 1000 random permutations for most tests (i.e., $N \geq 10$), except for several cases where the number of possible permutations was lower (when $N \leq 9$), in which case each unique permutation was used exactly once. To determine the presence of effects, wPLI$_Z$, dPAC$_Z$, and MI$_Z$ values at each frequency/frequency pair were compared to zero across patients (group) or data segments (individual); one-tailed tests were used because only above-zero effects are of interest. Comparisons between regions and directions were performed with two-tailed paired or unpaired tests as required. Clusters were deemed significant at $P < 0.05$ (one-tailed) and $P < 0.025$ (two-tailed); for wPLI$_Z$ we additionally show clusters with $P < 0.10$. For dPAC$_Z$ and MI$_Z$, clusters were required to span at least $2 \times 2$ frequency bins; no minimum cluster size was required for wPLI$_Z$. Circular distributions were tested using the Rayleigh test for uniformity (Fig. 3d, e). We further employed False Discovery Rate (FDR) correction for multiple comparisons[98] across clusters (Figs. 3c–e, 5c), using a $q$ of 0.05. In addition to group-level analyses, single-subject analyses indicated the presence of group effects across multiple patients (Supplementary Figs. 3, 6, 8).

**Reporting summary**. Further information on research design is available in the Nature Research Reporting Summary linked to this article.

## Data availability

Data are not publicly available due to privacy concerns related to clinical data, but data are available from the corresponding or senior author upon obtaining ethical approval.

## Code availability

All computer code used to analyze data is available from the corresponding author on request. Although no restrictions apply for code sharing, the interrelated nature of code and data prevent meaningful code usage without data access (see Data availability).

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

## Acknowledgements

This work was supported by the German Research Foundation (FE366/9-1 to J.F.) and a Wellcome Trust/Royal Society Sir Henry Dale Fellowship (107672/Z/15/Z to B.P.S.).

## Author contributions

Conceptualization by R.C. and J.F.; methodology by R.C., T.R., and J.F.; analysis by R.C.; interpretation by R.C., B.P.S., and J.F.; visualization by R.C. and T.R.; writing by R.C. and J.F.; funding acquisition by J.F.

## Competing interests

The authors declare no competing interests.
