## [Peer Review File · Communications Biology]

Reviewers' comments:

Reviewer #1 (Remarks to the Author):

This is a human study using (mainly) intracranial recordings during sleep (N2, N3, and REM) in both the temporal cortex and the hippocampus. Results showed phase synchrony between the hippocampus and the temporal cortex for spindles in both N2 and N3 sleep stages as well as for theta in N2 and REM sleep. Results also showed that hippocampal slow oscillation modulated the expression of faster activities (delta, theta, beta/gamma, and ripple) in the temporal cortex during N/3 sleep stage, but not in N2 or REM sleep. No modulations of higher frequencies in the hippocampus by the temporal cortex slow oscillation were found in any sleep stage.

However, my major concern is that authors refer to the temporal cortex as neocortex, like the neocortex is a uniform entity. The study, however important, was made specifically in the temporal cortex and results and conclusions should be drawn accordingly. As authors wrote: "While our patient sample offered much wider NC electrode coverage, we deliberately studied a relatively circumscribed NC area (Fig. 1, Table S2) to avoid confounding influences related to the non-homogeneous spatial expression of SOs and spindles 26,82–84." Slow oscillation is a travelling wave with most waves originating from the frontal cortex (Massimini et al., 2014). A lot of slow waves and spindles are local (Nir et al, 2011; Andrillon et al., 2011; Vyazovskiy et al, 2011). Slow and fast spindles also show different topographical distribution (De Gennaro and Ferrara, 2003). Therefore any study conducted to elucidate the coordination of activities between the neocortex and other structures should be very specific about the structures/areas investigated. While results of this study might be true for the temporal cortex, they might be completely wrong about other cortical areas.

Reviewer #2 (Remarks to the Author):

This manuscript by Cox et al. reports hippocampal-neocortical interactions during sleep in human. The paper is mainly methods-driven, and in my opinion the authors overstated their conclusion, particularly regarding the direction of communication between the hippocampus (HPC) and neocortex (NC).

Cross-frequency coupling (CFC) measures such as PAC and MI have been criticized, with concerns about spurious CFC estimates resulting from non-stationarities, sharp transients, non-uniform phase distributions, and other issues. This paper mainly uses dPAC and MI. What the authors found is that there is interaction between the HPC phase and NC amplitude, but not between HPC amplitude and NC phase. This however contrasts with most previous studies.

Specific comments:

1. Line 61: Hippocampal ripples were often defined between 100–300 Hz, not 60–100 Hz.
2. Did the authors record any hippocampal ripple activity (100–300 Hz)? In Figure 3, no coupling between hippocampal ripples and neocortical slow oscillation was observed. This contrasts with earlier studies reporting that hippocampal ripples occur at neocortical slow oscillation 'UP' state (in rodents). It is not clear how the current result may be reconciled with previous findings.
3. The authors should be consistent when they refer to the frequency band of slow oscillation (sometimes it was referred as 0.5–1 Hz, while at other times it was 0.5–1.5 Hz).

Response to reviewers

We very much appreciate and thank the reviewers for reading our manuscript, raising important points, and offering helpful suggestions. We have revised our manuscript to address their concerns. Below, we offer a point-by-point response to all comments. Line numbers refer to the *mainText_Revision1_ForReviewers* document (which differs from *mainText_Revision1_Clean* in terms of formatting).

Reviewer #1:

This is a human study using (mainly) intracranial recordings during sleep (N2, N3, and REM) in both the temporal cortex and the hippocampus. Results showed phase synchrony between the hippocampus and the temporal cortex for spindles in both N2 and N3 sleep stages as well as for theta in N2 and REM sleep. Results also showed that hippocampal slow oscillation modulated the expression of faster activities (delta, theta, beta/gamma, and ripple) in the temporal cortex during N/3 sleep stage, but not in N2 or REM sleep. No modulations of higher frequencies in the hippocampus by the temporal cortex slow oscillation were found in any sleep stage.

However, my major concern is that authors refer to the temporal cortex as neocortex, like the neocortex is a uniform entity. The study, however important, was made specifically in the temporal cortex and results and conclusions should be drawn accordingly. As authors wrote: "While our patient sample offered much wider NC electrode coverage, we deliberately studied a relatively circumscribed NC area (Fig. 1, Table S2) to avoid confounding influences related to the non-homogeneous spatial expression of SOs and spindles 26,82–84." Slow oscillation is a travelling wave with most waves originating from the frontal cortex (Massimini et al., 2014). A lot of slow waves and spindles are local (Nir et al, 2011; Andrillon et al., 2011; Vyazovskiy et al, 2011). Slow and fast spindles also show different topographical distribution (De Gennaro and Ferrara, 2003). Therefore any study conducted to elucidate the coordination of activities between the neocortex and other structures should be very specific about the structures/areas investigated. While results of this study might be true for the temporal cortex, they might be completely wrong about other cortical areas.

Response: We thank the reviewer for raising this important point, and we fully agree that the neocortex (NC) is far from homogenous, including in its expression of various sleep oscillations. It is therefore certainly possible that our observations are specific to the region studied. Correspondingly, we never intended to make strong claims that our results generalize to all of NC, but we can see how the original manuscript may have given that impression.

In fact, this issue intersects with one raised by Reviewer 2, who asked us to address inconsistencies with the literature in terms of HPC-NC directionality (see 2.5). Specifically, we believe our finding of HPC SOs affecting NC amplitudes but not vice versa, might have looked different when considering frontal cortex rich in high-amplitude SOs. At the same time, we do believe that the presence of local SOs and local spindles across most of NC increases the likelihood that similar hippocampal-cortical interactions can be found for other NC regions (in terms of frequencies involved, but not necessarily direction). We have altered our manuscript in several ways to better reflect our views and to address these concerns.

First, in the abstract we now make explicit that we considered lateral temporal NC:

Line 33: "Employing invasive electroencephalography in ten neurosurgical patients across a full night of sleep, we identified three broad classes of phase-based **communication between HPC and lateral temporal NC.**"

Second, the Introduction now mentions the non-uniform expression of sleep oscillations, while also pointing out local SOs and spindles across most of NC:

Line 101: "Here, we examined intracranial electrophysiological activity in a sample of 10 presurgical epilepsy patients during light NREM (N2), deep NREM (N3) and REM sleep. Specifically, we focused on HPC and lateral temporal cortex as a neocortical site relevant for long-term memory storage. **Although sleep oscillations are not expressed uniformly across NC^{61,62}, findings of local SOs and spindles in virtually all of NC³⁵ make lateral temporal cortex a suitable site for studying HPC-NC interactions.** We hypothesized that these areas exhibit interregional phase coordination, which could manifest in any or all of the three aforementioned forms of coupling."

Third, we revised our discussion at various places to better reflect the concerns regarding generalizability to all of NC:

Line 346: "**The lack of systematic NC-HPC PAC in our data may appear at odds with previous observations of NC-HPC SO-spindle²⁴, SO-ripple⁸⁵, and spindle-ripple coupling^{21,24,27}, which could be related to several factors. First, previous work often assessed NC activity with non-invasive scalp electrodes that aggregate activity over large spatial domains, thus reflecting common signals with relatively powerful drives. In contrast, the localized NC activity we considered here constitutes only a tiny fraction of all NC activity and may therefore exert a more limited influence on HPC activity (also see relative dissociation of scalp and NC signals in Supplementary Fig. 1). Second, and somewhat related, it is possible that modulation of HPC activity by NC SOs only becomes apparent when considering frontal regions with high SO densities, amplitudes, and synchrony (e.g.,^{62,85}). Indeed, interregional spindle synchrony is modulated by the phase of frontal, but not central or parietal, SOs⁴⁷, raising the possibility that SO activity recorded from lateral temporal areas is insufficient to impact HPC dynamics. We return to the issue of the NC recording site below.**"

Line 381: "**Our findings add to the debate on the directionality of HPC-NC dynamics during sleep.** Prior empirical evidence has been mixed, pointing towards HPC-NC^{66,85}, NC-HPC^{28,35,65}, or more elaborate bidirectional paths^{50,67,68}. Both our findings of HPC SOs coordinating NC faster activity but not *vice versa*, and of stronger modulations of theta phase synchrony by HPC SOs than NC SOs, are most consistent with the notion of HPC to NC directionality, as suggested by classical theoretical¹⁶ and computational⁸⁶ models. Based on these results, we propose that HPC SOs may influence plasticity not only within HPC but also in NC circuits⁶⁴. We emphasize, however, that observed directionality varies importantly with the precise electrophysiological phenomenon under consideration, **and as mentioned, may also depend on the particular NC regions examined.**"

Line 391: "We considered communication between HPC and lateral temporal cortex, a neocortical region important for long-term memory storage^{7,10,13,14}. More fundamentally, dominant sleep consolidation theories posit that the HPC-NC dialogue engages all NC areas involved in representing episodic memory components¹⁶⁻¹⁹, which for lateral temporal cortex could constitute higher-order visual, verbal, and semantic concepts^{8,9,11,12,15}. While our patient sample offered much wider NC electrode coverage, we deliberately studied a relatively circumscribed NC area (Fig. 1, Supplementary Table 2) to avoid confounding influences related to the non-homogeneous spatial expression

of SOs and spindles^{35,61,62,87}. Hence, we reiterate that the observed forms of communication, and their directionality in particular, may be specific to lateral temporal areas. At the same time, observations of local SOs and local spindles in widespread NC areas^{35,36,53} suggest that the reported forms of phase coordination may apply more broadly to other NC sites."

We hope that these revisions add important nuance and address the reviewer's concerns.

Reviewer #2:

This manuscript by Cox et al. reports hippocampal-neocortical interactions during sleep in human. The paper is mainly methods-driven, and in my opinion the authors overstated their conclusion, particularly regarding the direction of communication between the hippocampus (HPC) and neocortex (NC).

2.1 Cross-frequency coupling (CFC) measures such as PAC and MI have been criticized, with concerns about spurious CFC estimates resulting from non-stationarities, sharp transients, non-uniform phase distributions, and other issues. This paper mainly uses dPAC and MI.

Response: We fully agree with the reviewer that cross-frequency metrics are not perfect. However, in our view this is true for virtually all metrics, each offering different advantages and drawbacks. Hence, we sought to use the most appropriate metrics and analysis strategies given our study goals, while remaining aware of their limitations. We now address the specific issues listed by the reviewer.

First, we used a measure of phase-amplitude coupling that included a "debias" term to handle non-uniform phase distributions (dPAC). In our view, this is a particularly important issue given the non-sinusoid waveform of SOs with its sharper down than up state (and potentially similar issues for other frequencies). Importantly, extensive simulation work has shown this approach to be effective (van Driel et al., 2015). We have revised the Methods to more clearly indicate that this metric handles non-uniform phase distributions:

Line 514: "We employed an adaptation of the mean vector length method⁹⁴ that adjusts for possible bias stemming from non-sinusoidal shapes of *f1* and associated non-uniform phase distributions⁶⁷."

Second, regarding non-stationarities, we emphasize our use of surrogates, whereby any non-stationarities affecting the data are equally present for the surrogates, and should therefore be accounted for. This holds equally for all metrics (wPLI, dPAC, and MI). To improve the manuscript's organization, we have added a new *Surrogate Construction* paragraph, containing all the surrogate details that were previously contained in the paragraphs for individual metrics:

Line 548: "**Surrogate construction**

For phase synchrony, surrogates were constructed per 60 s segment and frequency by repeatedly (n = 100) time shifting the phase time series of the seed channel by a random amount between 1 and 59 s, and recalculating wPLI for each iteration. Similarly, for cross-frequency phase-amplitude coupling, we constructed a surrogate distribution of coupling strengths per 60 s segment, frequency pair, and same/cross-site condition, by repeatedly (n = 100) time shifting the *f1* phase time series with respect to the *f2* amplitude time series, recalculating the mean vector length (dPAC) for each iteration. For the cross-frequency modulation of phase synchrony, surrogate

distributions were constructed by repeatedly (n=100) shuffling the pairing of *f1* and *f2* 60 s phase segments (disallowing pairings where individual segments were unaltered), and recalculating MI across segments for each iteration, rather than per segment. Since the number of unique segment pairings depends on the number of available segments, 2 patients were excluded from N3 analyses.

Note that time shifting accounts for non-stationarities in the data, and is a more conservative approach than fully scrambling time series, which may result in spurious effects⁹⁵. For each metric, the surrogate distribution was used to z-score the raw values. Thus, the z-scored measures ($wPLI_z$, $dPAC_z$, MI_z) indicate how far, in terms of standard deviations, the observed coupling estimate is removed from the average coupling estimate under the null hypothesis of no coupling."

Third, we agree that most PAC metrics (including ours) are sensitive to aspects of waveform shape (including, but not limited to, transients), and we cannot entirely rule out their contribution to our results. That said, the observed forms of local PAC, as reported in the supplementary materials, are very much in line with established interactions between different electrophysiological phenomena (e.g., SOs, sharp waves, spindles, ripples). This instills confidence that analyses of interregional PAC also capture true oscillatory interactions. We note that we are aware of recent suggestions to consider waveform shape in detail (Cole and Voytek, 2017). However, such approaches are not feasible when evaluating coupling for a large number of frequency pairs, as we did here.

Finally, just for clarity we would like to note that our MI metric is not a measure of phase-amplitude coupling, but a measure of "low-frequency phase to high-frequency phase synchronization" coupling.

Given that we deem the transient/asymmetrical oscillation the most relevant concern, we now include this point in the Discussion:

Line 412: "Nonetheless, PAC metrics capture both true oscillatory interactions and features related to waveform shape⁸⁸, which may have contributed to our results. Although we believe that the specific patterns of results (e.g., asymmetrical HPC-NC modulation limited to the SO-band) are most parsimoniously explained by interacting oscillations, future work should scrutinize individual waveforms to fully understand the origin of each of the observed effects."

2.2 What the authors found is that there is interaction between the HPC phase and NC amplitude, but not between HPC amplitude and NC phase. This however contrasts with most previous studies.

Response: Please see our response to 2.5.

Specific comments:

2.3 ~~Line 61:~~ Hippocampal ripples were often defined between 100–300 Hz, not 60–100 Hz.

Response: Thank you for raising this issue. This discrepancy is entirely due to species differences, with human ripples having now repeatedly been shown to be much slower than their rodent counterparts (please see, for instance, refs. 20-26). As we realize this will not be apparent to all readers, we now include a statement in the introduction to clarify:

Line 59: "Neural oscillations, especially non-rapid eye movement (NREM) neocortical slow oscillations (SOs; 0.5–1.5 Hz), thalamocortical sleep spindles (12.5–16 Hz), and **hippocampal ripples (60–100 Hz^{20–26}; note that human ripples are substantially slower than ~150–250 Hz rodent ripples^{27,28}**), are widely held to mediate this HPC-NC memory transfer and consolidation process^{29–34}, particularly given the presence of both SOs and spindles in HPC^{22,24,35–37}."

2.4 2. *Did the authors record any hippocampal ripple activity (100–300 Hz)?*

Response: As we evaluated frequencies from 0.5 Hz up to 200 Hz, it was possible to detect effects >100 Hz. Indeed, we sometimes observed single-patient instances of SO-ripple or delta-ripple PAC where ripple clusters extended up to 200 Hz (e.g., Supplementary Fig. 6A, N3, p7 and p9). However, in each of these cases, strongest modulation was around 80 Hz, consistent with the aforementioned notion that ripples are slower in humans.

2.5 *In Figure 3, no coupling between hippocampal ripples and neocortical slow oscillation was observed. This contrasts with earlier studies reporting that hippocampal ripples occur at neocortical slow oscillation 'UP' state (in rodents). It is not clear how the current result may be reconciled with previous findings.*

Response: Thank you for addressing this important point. First, we would like to note that we already tackled a very similar concern in the original Discussion, in relation to a) NC-HPC SO-spindle coupling, and b) NC-HPC spindle-ripple coupling (both of which we did not observe). We had failed to consider the additional absence of c) NC-HPC SO-ripple coupling, as suggested by the reviewer. We now discuss these three effects comprehensively, providing a potential explanation by considering that our NC recording site (lateral temporal) may not be optimal for finding NC-HPC SO-spindle or SO-ripple modulations. (See also our response to Reviewer 1.)

Line 347: "**The lack of systematic NC-HPC PAC in our data may appear at odds with previous observations of NC-HPC SO-spindle²⁴, SO-ripple⁸⁵, and spindle-ripple coupling^{21,24,27}, which could be related to several factors. First, previous work often assessed NC activity with non-invasive scalp electrodes that aggregate activity over large spatial domains, thus reflecting common signals with relatively powerful drives. In contrast, the localized NC activity we considered here constitutes only a tiny fraction of all NC activity and may therefore exert a more limited influence on HPC activity (also see relative dissociation of scalp and NC signals in Supplementary Fig. 1). Second, and somewhat related, it is possible that modulation of HPC activity by NC SOs only becomes apparent when considering frontal regions with high SO densities, amplitudes, and synchrony (e.g.,^{62,85}). Indeed, interregional spindle synchrony is modulated by the phase of frontal, but not central or parietal, SOs⁴⁷, raising the possibility that SO activity recorded from lateral temporal areas is insufficient to impact HPC dynamics. We return to the issue of the NC recording site below.**"

2.6 3. *The authors should be consistent when they refer to the frequency band of slow oscillation (sometimes it was referred as 0.5–1 Hz, while at other times it was 0.5–1.5 Hz).*

Response: Thank you for pointing out this inconsistency. We now label the SO-band as 0.5-1.5 Hz throughout the manuscript.

We hope that these responses address all remaining concerns.

References

Cole, S. R. & Voytek, B. Brain Oscillations and the Importance of Waveform Shape. *Trends Cogn. Sci.* **21**, 137–149 (2017).

van Driel, J., Cox, R. & Cohen, M. X. Phase-clustering bias in phase–amplitude cross-frequency coupling and its removal. *J. Neurosci. Methods* **254**, 60–72 (2015).

REVIEWERS' COMMENTS:

Reviewer #1 (Remarks to the Author):

Authors addressed all my concerns in this revised manuscript.

Reviewer #2 (Remarks to the Author):

The authors have appropriately addressed my concerns. I have no further comments.